# Comprehensive genomic characterization of HER2-low and HER2-0 breast cancer

Paolo Tarantino [1,2,3,4,13] ✉, Hersh Gupta[1,5,13], Melissa E. Hughes[1], Janet Files[1], Sarah Strauss[1], Gregory Kirkner[1], Anne-Marie Feeney[1], Yvonne Li[1], Ana C. Garrido-Castro [1,2,3], Romualdo Barroso-Sousa[6,7], Brittany L. Bychkovsky [2,3,8], Simona DiLascio[1], Lynette Sholl [1], Laura MacConaill[1], Neal Lindeman[1,12], Bruce E. Johnson[1], Matthew Meyerson [1,3,5], Rinath Jeselsohn [1,2,3], Xintao Qiu [1], Rong Li [1], Henry Long [1], Eric P. Winer[9], Deborah Dillon[3,10], Giuseppe Curigliano [11], Andrew D. Cherniack[1,3,5], Sara M. Tolaney [1,2,3,14] & Nancy U. Lin[1,2,3,14]

The molecular underpinnings of HER2-low and HER2-0 (IHC 0) breast tumors remain poorly defined. Using genomic findings from 1039 patients with HER2-negative metastatic breast cancer undergoing next-generation sequencing from 7/2013-12/2020, we compare results between HER2-low ($n$ = 487, 47%) and HER2-0 tumors ($n$ = 552, 53%). A significantly higher number of *ERBB2* alleles (median copy count: 2.05) are observed among HER2-low tumors compared to HER2-0 (median copy count: 1.79; $P$ = 2.36e-6), with HER2-0 tumors harboring a higher rate of *ERBB2* hemideletions (31.1% vs. 14.5%). No other genomic alteration reaches significance after accounting for multiple hypothesis testing, and no significant differences in tumor mutational burden are observed between HER2-low and HER2-0 tumors (median: 7.26 mutations/megabase vs. 7.60 mutations/megabase, $p$ = 0.24). Here, we show that the genomic landscape of HER2-low and HER2-0 tumors does not differ significantly, apart from a higher *ERBB2* copy count among HER2-low tumors, and a higher rate of *ERBB2* hemideletions in HER2-0 tumors.

For over two decades, HER2 has been classified in a binary fashion in breast oncology: positive, if overexpressed (immunohistochemistry [IHC] 3+) or amplified (positive in situ hybridization [ISH]), and negative in the absence of these alterations[1]. Based on this paradigm, 80–85% of breast cancers were traditionally defined as HER2-negative, despite over half of these having detectable HER2 protein by IHC[2]. According to the latest American Society of Clinical Oncology/College of American

Pathologists (ASCO/CAP) Guidelines for HER2 testing, HER2-negative breast cancers include tumors with HER2 IHC scores of 0, 1+ or 2+/ISH-negative[1], for which treatment with traditional HER2-targeting agents has not demonstrated meaningful clinical benefits[3,4]. A paradigm shift in this setting has however occurred with the emergence of anti-HER2 antibody-drug conjugates (ADCs)[5], several of which have demonstrated relevant activity among patients with HER2-negative

[1]Medical Oncology, Dana-Farber Cancer Institute, Boston, MA, USA. [2]Breast Oncology Program, Dana-Farber Brigham Cancer Center, Boston, MA, USA. [3]Harvard Medical School, Boston, MA, USA. [4]Department of Oncology and Hematology-Oncology, University of Milano, Milano, Italy. [5]Broad Institute of Harvard and MIT, Cambridge, MA, USA. [6]Dasa Institute for Education and Research (IEPD), Brasilia, Brazil. [7]Dasa Oncology/Hospital Brasilia, Brasilia, Brazil. [8]Division of Cancer Genetics and Prevention, Dana-Farber Cancer Institute, Boston, MA, USA. [9]Yale Cancer Center, Yale School of Medicine, Smilow Cancer Hospital, New Haven, CT, USA. [10]Department of Pathology, Brigham and Women's Hospital, Boston, MA, USA. [11]Division of Early Drug Development, European Institute of Oncology IRCCS, Milano, Italy. [12]Present address: Department of Pathology, Weill Cornell Medicine, New York, NY, USA. [13]These authors contributed equally: Paolo Tarantino, Hersh Gupta. [14]These authors jointly supervised this work: Sara M. Tolaney, Nancy U. Lin. ✉e-mail: paolo_tarantino@dfci.harvard.edu

metastatic breast cancer (MBC) exhibiting HER2-low expression, defined as IHC 1+ or 2+/ISH-negative[6–10]. In this setting, the phase 3 DESTINY-Breast04 trial demonstrated that trastuzumab deruxtecan (T-DXd) is associated with substantial improvements in progression-free survival and overall survival compared with traditional chemotherapy among patients with HER2-low MBC[11], leading to the approval of the drug by the U.S. Food and Drug Administration and European Medicines Agency, and reshaping treatment algorithms in breast oncology.

Currently, patients with HER2-low MBC may be offered T-DXd, whereas T-DXd is not considered the standard of care in patients with HER2 IHC 0 (hereafter referred to as HER2-0) MBC. However, it remains unclear whether HER2-low breast cancer should be considered a distinct molecular entity, with different genomic underpinnings compared with HER2-0 tumorsbreast cancer[12,13].

In this study, we aim to characterize the genomic profile of HER2-low tumors across a large population of patients with MBC evaluated at Dana-Farber Cancer Institute and to compare it with the genomic profile of HER2-0 tumors after correcting for potential confounding variables.

## Results

### Cohort characteristics

A total of 1039 patients met the inclusion criteria and were included in this study (Supplementary Fig. 1). Baseline demographics and clinicopathologic characteristics according to HER2 status are reported in Table 1, Supplementary Table 1, and in the Source Data. In total, 487 patients had HER2-low (46.9%), and 552 patients had HER2-0 (53.1%) status in the tumor tested with NGS. Most of the tumors were from metastatic lesions ($n = 777$, 74.8%), and only a minority were primary tumors ($n = 238$, 22.9%) or local recurrences ($n = 24$, 2.2%). In terms of procedures utilized to collect tissue, most of the tissue derived from tumor biopsies performed in clinical practice for the biologic characterization of tumors (72%, $n = 750$), with a similar proportion of samples collected via biopsy across HER2-low (71%, $n = 344$) and HER2-0 patients (73%, $n = 406$). Among the samples collected in the metastatic setting, the median time from metastatic diagnosis to tissue collection was 12 days (interquartile range: 0–153).

Overall, 706 (67.9%) of the patients had estrogen receptor (ER)-positive, 39 (3.8%) had ER-low, and 288 (27.7%) had ER-negative disease; 6 (0.6%) did not have an available ER status on the specimen undergoing NGS. A significant difference was observed in the distribution of ER status between HER2-low and HER2-0 samples: 370 (76.0%) HER2-low vs. 336 (60.9%) HER2-0 ER-positive samples, 18 (3.7%) vs. 21 (3.8%) ER-low, and 94 (19.3%) vs. 194 (35.1%) ER-negative samples ($P = 1.29e\text{-}7$). As would be expected given the differences in ER status between patients with HER2-low and HER2-0 tumors, further differences were observed in the time from initial diagnosis to NGS testing, the incidence of brain metastases, the incidence of liver metastases and the receipt of prior endocrine treatments (all $P < 0.05$).

### Genomic landscape of HER2-low and HER2-0 breast cancers

Among HER2-low tumors, the most common oncogenic variants were identified in *TP53* (34.8% in the overall HER2-low cohort; 79.8% ER-negative, 45.5% ER-low, 22.6% ER-positive), *PIK3CA* (32.9%; 14.9%, 22.7%, 38.2%), *CDH1* (15.3%; 5.3%, 22.7%, 17.5%), *GATA3* (13.5%; 2.1%, 13.6%, 16.1%), and *ESR1* (10.8%; 0%, 9.1%, 13.4%) (Figs. 1 and 2). Oncogenic variants in the same five genes were also found to be the most common in the HER2-0 cohort: *TP53* (50.3% whole HER2-0 cohort; 87.2% ER-negative, 81.8% ER-low, 26.5% ER-positive), *PIK3CA* (32.2%; 11.8%, 36.4%, 43.8%), *CDH1* (11.8%; 3.1%, 18.2%, 16.4%), *GATA3* (8.0%; 1.0%, 0.0%, 12.5%), and *ESR1* (7.1%; 0.0%, 0.0%, 11.6%) (Figs. 1 and 2).

In both the HER2-low and the HER2-0 cohorts, only 3 oncogenic copy number variations (CNVs) were present in more than 5% of cases: high amplifications of *CCND1* (13.3% HER2-low, 13.2% HER2-0), high amplifications of *FGFR1* (12.3% HER2-low, 9.8% HER2-0), and high

**Table 1 | Patient and tumor characteristics among patients with metastatic breast cancer included in the study**

| | Total population ($n = 1039$) | Patients with HER2-low tumors tested ($n = 487$) | Patients with HER2-0 tumors tested ($n = 552$) | P-value |
|---|---|---|---|---|
| **Age in years**, median (min, max) | 55 (22, 89) | 56 (25, 89) | 54 (22, 88) | 0.06 |
| **Sex**, n (%) | – | – | – | 0.92 |
| Female | 1028 (99) | 482 (99) | 546 (99) | – |
| Male | 11 (1) | 5 (1) | 6 (1) | – |
| **Ethnicity**, n (%) | – | – | – | 0.14 |
| African American/Black | 50 (4.8) | 20 (4.1) | 30 (5.4) | – |
| Asian or Pacific Islander | 39 (3.7) | 16 (3.3) | 23 (4.2) | – |
| White | 897 (86.3) | 433 (88.9) | 465 (84.1) | – |
| Other | 21 (2.0) | 9 (1.8) | 12 (2.2) | – |
| Unknown | 32 (3.1) | 9 (1.8) | 23 (4.2) | – |
| **Ethnicity**, n (%) | – | – | – | 0.69 |
| Hispanic | 41 (3.9) | 18 (3.7) | 23 (4.2) | – |
| Non-Hispanic | 936 (90.1) | 437 (89.7) | 499 (90.4) | – |
| Unknown | 62 (6.0) | 32 (6.6) | 30 (5.4) | – |
| **Stage at initial diagnosis**, n (%) | – | – | – | 0.71 |
| DCIS | 17 (1.6) | 8 (1.6) | 9 (1.6) | – |
| I | 161 (15.5) | 78 (16.0) | 83 (15.0) | – |
| II | 339 (32.6) | 153 (31.4) | 186 (33.7) | – |
| III | 266 (25.6) | 118 (24.2) | 148 (26.8) | – |
| IV | 245 (23.6) | 124 (25.5) | 121 (21.9) | – |
| Unknown | 11 (1.1) | 6 (1.2) | 5 (0.9) | – |
| **Hormone receptor status of sample tested**, n (%) | – | – | – | – |
| Positive | 761 (73.2) | 392 (80.5) | 369 (66.9) | <0.0001 |
| Negative | 272 (26.2) | 90 (18.5) | 182 (33.0) | – |
| Not done | 6 (0.6) | 5 (1.0) | 1 (0.1) | – |
| **Estrogen receptor status of sample tested**, n (%) | – | – | – | <0.0001 |
| Positive | 706 (67.9) | 370 (76.0) | 336 (60.9) | – |
| Positive low | 39 (3.8) | 18 (3.7) | 21 (3.8) | – |
| Negative | 288 (27.7) | 94 (19.3) | 194 (35.1) | – |
| Not done | 6 (0.6) | 5 (1.0) | 1 (0.2) | – |
| **Histology at initial diagnosis**, n (%) | – | – | – | 0.09 |
| DCIS | 17 (1.6) | 8 (1.6) | 9 (1.6) | – |
| Invasive ductal carcinoma (IDC) | 758 (73.0) | 338 (69.4) | 420 (76.1) | – |
| Invasive lobular carcinoma (ILC) | 135 (13.0) | 70 (14.4) | 65 (11.8) | – |
| Mixed (IDC & ILC) | 93 (8.9) | 47 (9.7) | 46 (8.3) | – |
| Other | 11 (1.1) | 5 (1.0) | 6 (1.1) | – |
| Unknown | 25 (2.4) | 19 (3.9) | 6 (1.1) | – |
| **Type of specimen tested**, n (%) | – | – | – | 0.40 |
| Primary breast | 238 (22.9) | 113 (23.2) | 125 (22.6) | – |
| Local recurrence | 24 (2.3) | 8 (1.6) | 16 (2.9) | – |
| Metastasis | 777 (74.8) | 366 (75.2) | 411 (74.5) | – |
| **OncoPanel version**, n (%) | – | – | – | 0.90 |
| V1 | 62 (6.0%) | 29 (6.0%) | 33 (6.0%) | – |
| V2 | 268 (25.8%) | 128 (26.3%) | 140 (25.4%) | – |
| V3 | 709 (68.2%) | 330 (67.8%) | 379 (68.7%) | – |
| **Time from initial met diagnosis to OncoPanel test (median, min, max)**[a] | 12 (0, 4224) | 15.5 (0, 4224) | 9 (0, 3530) | 2E-04 |

All tests were carried out using two-sided tests. For comparisons with categorical variables, chi-square test was used, while continuous or nominal variables were compared using a t-test. Duration to OncoPanel test was calculated using a Wilcoxon test. The test statistic is the underlying distribution for each of these tests with normally calculated degrees of freedom.

*HER2* human epidermal growth factor receptor 2, *DCIS* ductal carcinoma in situ.

[a]Excludes those with Procedure Dates Before Metastatic Diagnosis.

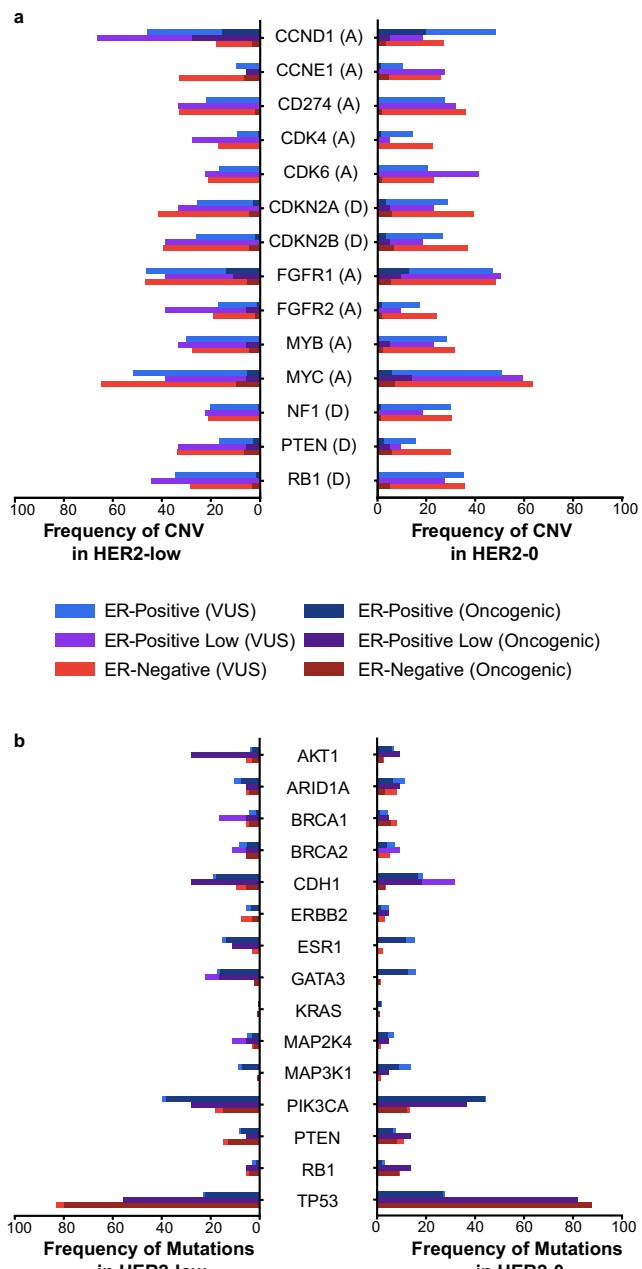

**a**

**b**

**Fig. 1 | Frequency of most common genomic alterations by HER2 status. a** The frequency of copy number variations (CNV); **b** the frequency of gene mutations. Shading represents the percentage of oncogenic events. An annotation of "(A)" beside a gene represents high amplification and "(D)" represents a deep or 2-copy deletion as the oncogenic event. Unshaded bars represent any other copy number event. HER2 human epidermal growth factor receptor 2, ER estrogen receptor.

status. Common mutations (present in >4% of either HER2-0 or HER2-low samples) found to be enriched among HER2-low samples included *MTOR* mutations (Odds Ratio [OR], 1.95; 95% CI, 1.04–3.64; $P = 0.04$), while those enriched among HER2-0 included *MAP3K1* mutations (OR, 0.59; 95% CI, 0.40–1.00, $P = 0.03$), oncogenic *NF1* mutations (OR, 0.48; 95% CI: 0.24–0.97; $P = 0.04$), and oncogenic *TP53* mutations (OR, 0.72; 95% CI, 0.53–0.97; $P = 0.03$). However, when adjusting for multiple hypothesis testing, no difference in the distribution of mutations was found between HER2-low and HER2-0 tumors ($q = 0.96$ for mentioned genes, Fig. 3 only including common mutations for simpler visualization).

When comparing oncogenic CNVs, only ESR1 (OR: 4.89 [1.02–23.13], $P = 0.05$) and IGF1R (OR: 2.53 [1.00–6.40], $P = 0.05$) amplifications were enriched in HER2-low tumors, whereas no oncogenic copy number alterations were enriched in HER2-0 compared to HER2-low tumors. As CNVs are less common, all oncogenic CNVs are highlighted in the text as only including common CNVs would limit the number severely. When adjusting for multiple hypothesis testing, however, no difference in the incidence of CNVs retained significance (Fig. 3). A full table with all converged logistic regression model for all mutations and all CNVs is provided in the Supplementary Data File.

### Tumor mutational burden

No significant differences in estimated tumor mutational burden (TMB) were observed according to HER2-low status, with a median TMB of 7.26 (0.76–85.94) for HER2-low and 7.60 (0.00–111.36) for HER2-0 samples ($P = 0.24$; stratified $P$ by ER status = 0.28, stratified Kruskal-Wallis test). Overall, 7.1% of the HER2-0 and 6.6% of the HER2-low tumors were found to be hypermutated ($P = 0.86$).

### *ERBB2* copy count

A total of 786 samples were found adequate for *ERBB2* copy counts estimation (HER2-low=369, HER2-0 = 417). Among HER2-low samples, 56 (15.2%) had a single-copy deletion of *ERBB2*, 247 (66.9%) had no change, and 66 (17.9%) had an allelic gain, versus 128 (30.7%), 252 (60.4%), and 37 (8.9%) for HER2-0, respectively (Fig. 4), resulting in a statistically significant difference in distribution after accounting for ER status (HER2-low median copy count: 2.05, HER2-0 median copy count: 1.79, $P = 2.36e-6$, Cochran-Mantel-Haenszel [CMH] test using ER status as the strata). When comparing HER2 expression against IHC, similar results are returned with control for ER status (IHC 0 median copy count: 1.79, IHC 1+ median copy count: 2.02, IHC 2+ median copy count: 2.08, $P = 4.11e-6$, CMH test using ER status as the strata) (Fig. 4).

### Sensitivity analysis of metastatic samples

Overall, 801 patients with samples collected while having metastatic disease and having a known ER status were included in the sensitivity analysis (HER2-low = 385, HER2-0 = 416).

Consistent with the overall analysis, HER2-low samples were more likely to be ER-positive (78.7%, vs. 63.2% in HER2-0, $P = 2.87e-6$). The top 5 mutated genes for both HER2-low and HER2-0 subgroups were *TP53, PIK3CA, CDH1, ESR1*, and *GATA3* (Supplementary Fig. 2), with comparable incidence compared to the overall cohort. CNVs also followed a pattern consistent with the overall cohort, with amplifications in *CCND1, FGFR1*, and *MYC* being the most common alterations (Supplementary Fig. 2). When comparing the mutational landscape and the frequency of CNV based on HER2-low expression, no significant differences were found after adjusting for multiple hypothesis testing (Supplementary Fig. 3). TMB also did not differ significantly, with HER2-low tumors having a median TMB of 7.26 (0.76–85.94), vs. 7.60 (1.33–53.99) for HER2-0 tumors ($P = 0.07$, stratified $P = 0.10$).

When comparing the estimated *ERBB2* genomic copy count, both HER2-low status and IHC expression were significantly associated with *ERBB2* copy count. A total of 575 samples (HER2-low=291, HER2-zero=284) were used in this specific analysis. Among HER2-low

amplifications of *MYC* (6.1% HER2-low, 6.1% HER2-0) (Figs. 1 and 2). As any CNV event that were not high amplifications for oncogenes and 2-copy deletions for tumor suppressor genes were included in the "variants of unknown significance (VUS)" category, a significantly higher number of VUSs appear in Fig. 1 as compared to other studies. Other commonly altered genes are shown in Figs. 1 and 2. Of note, no homozygous loss of *ERBB2* was observed in this cohort.

### Comparison of genomic profile by HER2 status

We next compared the genomic landscape of HER2-low and HER2-0 tumors. Logistic regression was used to account for background mutation or copy number alteration rate, as well as to adjust for ER

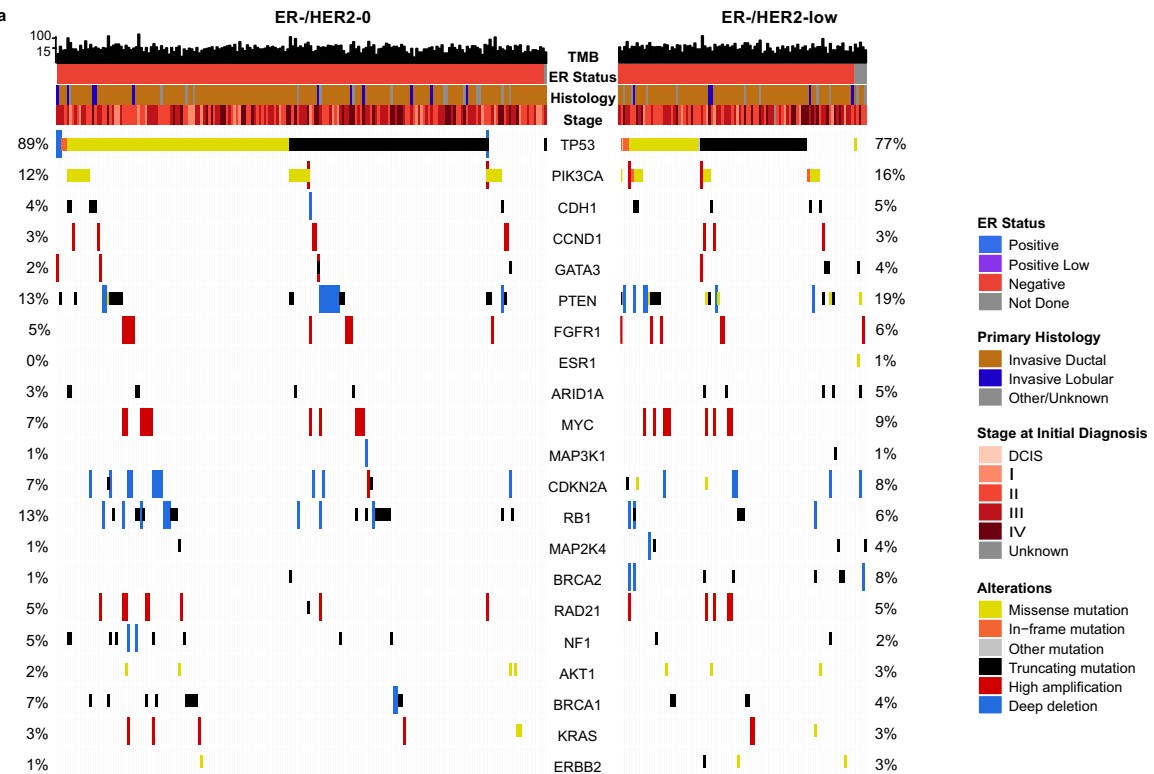

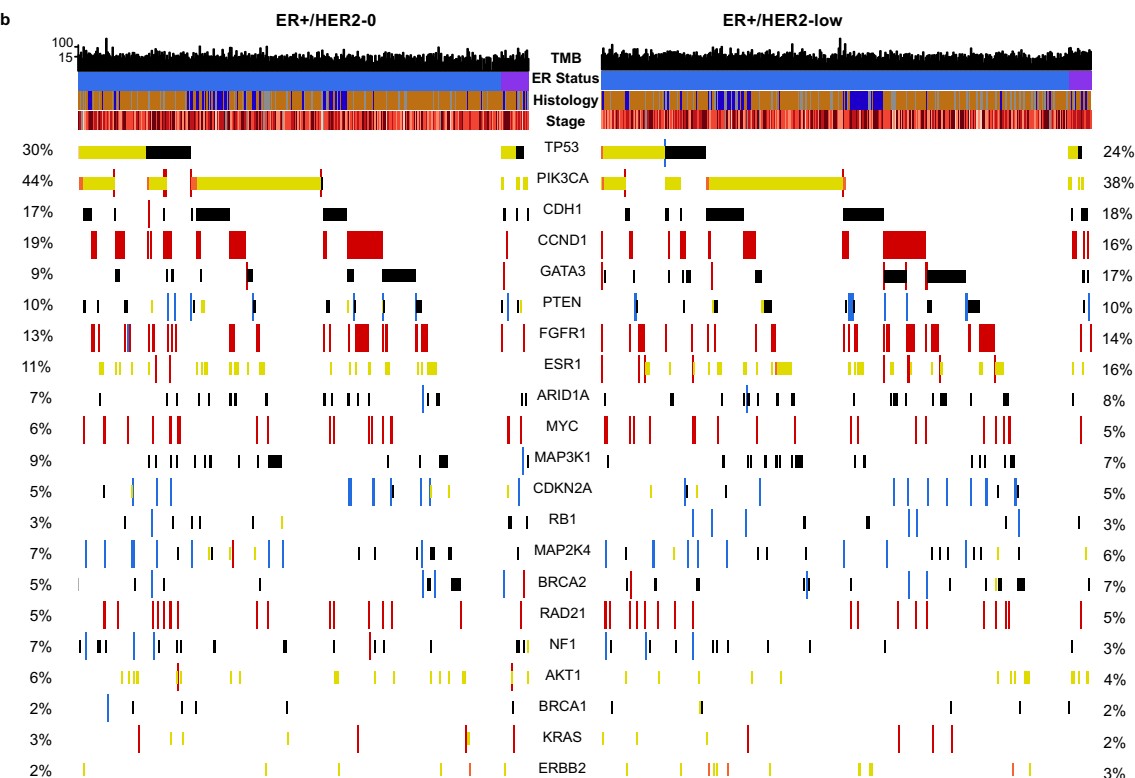

**Fig. 2 | OncoPrint view of the genomic landscape of HER2-0 (*n* = 552) and HER2-low (*n* = 487) metastatic breast cancer. a** The OncoPrint of ER-negative tumors, divided by HER2 status; **b** the OncoPrint of ER-positive tumors, divided by HER2 status. Genes are ordered by frequency of variants in the overall study population. Percentages listed show the frequency of alterations in HER2-0 and HER2-low, respectively. All variants represent oncogenic mutations or deep deletions/high amplifications. TMB (mut/mb) is recorded on the top of the plot. HER2 human epidermal growth factor receptor 2, MBC metastatic breast cancer, ER estrogen receptor, TMB tumor mutational burden.

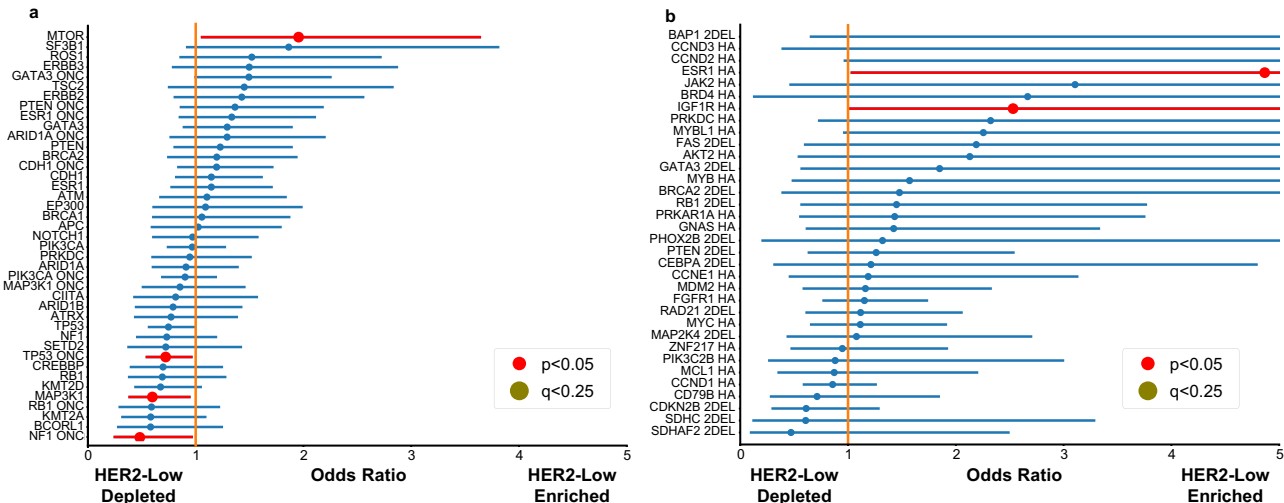

Fig. 3 | Enrichment analysis of genomic alterations between HER2-low (*n* = 482) and HER2-0 (*n* = 551). a The enrichment analysis for mutations; b the enrichment analysis for copy number variations (CNV). Modeling was done using multivariate logistic regression accounting for ER status and background rate of either mutation or copy number events, using the statsmodel package in Python. ER-low cases were included in the ER-positive group. Only models that reached a significant value for rejecting the log-likelihood null were included after multiple hypothesis correction using BH-FDR, as well as those that converged after 500 iterations. Only mutations that appeared in over 4% of either all HER2-0 or HER2-low samples were included. On the left, lines labeled "_ONC" represent only oncogenic mutations, while the

CNVs were done on 2DELs or high amplifications for tumor suppressor genes and oncogenes, respectively (labeled in the figure). Error bars are reported as the 95% confidence interval. *P*-values are determined as the likelihood of the model's calculated coefficients under the assumption that the true coefficients are 0 and are reported as two-tailed. Multiple hypothesis correction was done using BH-FDR. Exact *p*-values are reported in the source data of this figure. These are the same statistical tests used in the Supplementary Data File, Supplementary Figs. 3 and 6. CNV copy number variations, ER estrogen receptor, HER2 human epidermal growth receptor factor.

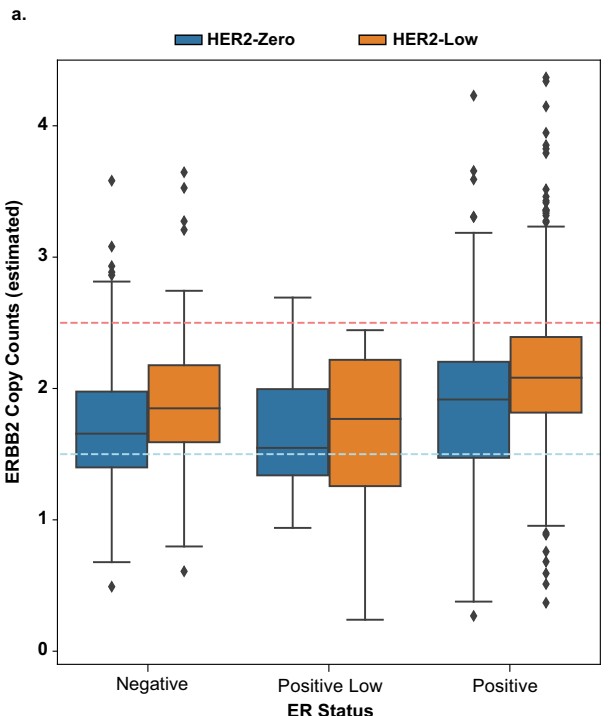

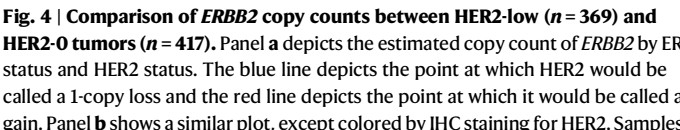

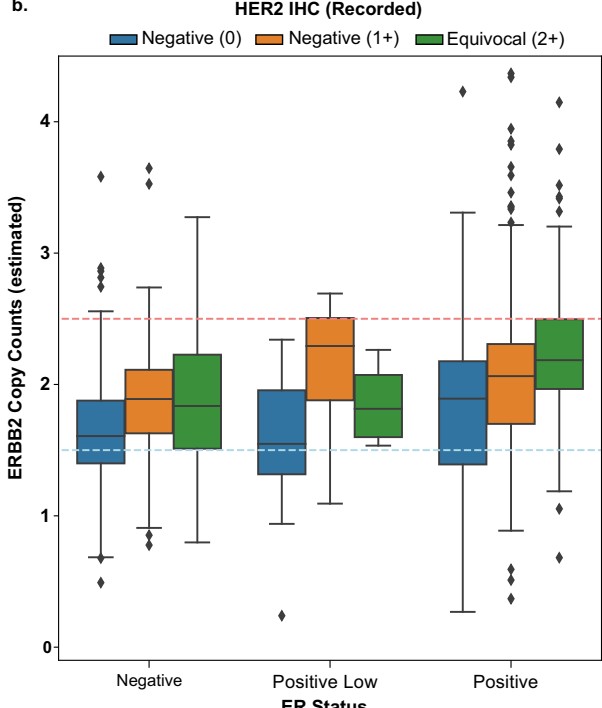

Fig. 4 | Comparison of *ERBB2* copy counts between HER2-low (*n* = 369) and HER2-0 tumors (*n* = 417). Panel a depicts the estimated copy count of *ERBB2* by ER status and HER2 status. The blue line depicts the point at which HER2 would be called a 1-copy loss and the red line depicts the point at which it would be called a gain. Panel b shows a similar plot, except colored by IHC staining for HER2. Samples

with a recorded IHC of unspecified value were excluded (HER2-low = 369 and HER2-0 = 417). Box plots are constructed with the central line as the median, the outer lines of the box as the lower and upper quartile, and whiskers are equal to 1.5x the closest quartile. HER2 human epidermal growth factor receptor 2, ER estrogen receptor, IHC immunohistochemistry.

samples, 13.0% had a single-copy deletion, 68.7% with no change, and 18.3% with copy number gain (median copy count: 2.05), while among HER2-0 samples, 32.3% had single-copy deletion, 60.8% no change, and 6.9% copy number gain (median copy count: 1.76), mirroring the results observed in the overall cohort. Both HER2-low status and the HER2 IHC score were found significantly associated with the estimated *ERBB2* copy count when corrected for ER status ($P = 1.42e-7$ and $P = 5.53e-6$, respectively $P$) (Supplementary Fig. 4).

### Sensitivity analysis of IHC 2+ vs. IHC 0 samples

Given some studies suggesting incomplete concordance in the pathologic scoring of HER2 IHC 0 vs 1+, a sensitivity analysis was carried out by comparing only HER2-low designated samples that stained IHC 2+/not amplified against the HER2-0 cohort for samples with a recorded ER status. This filtering left a total of 759 samples (549 HER2-0, 210 HER2-low) for analysis.

Slightly more HER2 IHC 2+ samples were ER-positive (78.1%, $P = 5.86e-6 < 0.001$) in this smaller cohort, while TMB was not significantly different between the two groups, with a median TMB of 7.26 (0.76–38.02) for HER2 2+ and a median TMB of 7.60 for HER2-0 (0.0–111.36, $P = 0.86$). Common oncogenic genomic mutations in the HER2 IHC 2+ cohort included the same as in the larger analysis with *PIK3CA* (36.2% HER2 IHC 2+ samples; 26.3% ER-negative, 37.5% ER positive-low, 38.7% ER-positive), *TP53* (32.9%; 78.9%, 87.5%, 19.6%), *CDH1* (14.8%; 7.9%, 12.5%, 16.6%), *GATA3* (13.8%; 0%, 0%, 17.8%), and *ESR1* (11.4%; 0%, 12.5%, 14.1%) the most common, which was similar to the overall cohort, though *TP53* was replaced by *PIK3CA* as the most common mutation. Copy number events were also similar and a plot of commonly altered samples can be seen in Supplementary Fig. 5.

When comparing the mutational landscape and copy number landscape, no genes rose to significance when accounting for multiple hypothesis testing (Supplementary Figs. 5 and 6). Mutations in similar genes, *TP53* and *NF1*, were found to be enriched in HER2-0 before multiple hypothesis testing. As the results of comparing the HER2 IHC and *ERBB2* copy count values were already described, the values are not restated here.

## Discussion

The demonstration of the targetability of HER2-low expression in breast oncology has ignited an extensive debate regarding whether to consider HER2-low a distinct entity within HER2-negative breast cancers. Among the criteria required to support the definition of a new entity, a critical one pertains to the presence of distinct patterns of genomic alterations[14]. Key genomic and gene expression differences are indeed established among traditional subtypes of breast cancer (i.e., luminal, HER2-positive, triple-negative)[14]. However, in our large study, including over 1000 patients with metastatic breast cancer undergoing genomic sequencing, after accounting for the confounding factor of ER expression, we did not identify any significant difference in gene mutations, CNV, nor in TMB between HER2-low and HER2-0 breast tumors, except for a higher *ERBB2* copy count expectedly found in HER2-low tumors. These results are concordant with those of other recent genomic studies, highlighting marginal differences in the genomic landscape between HER2-low and HER2-0 tumors, after correcting for ER expression[15,16]. For instance, in a cohort of 3608 HER2-negative patients subjected to targeted NGS sequencing, Marra et al. found no difference in mutational signatures and tumor mutational burden overall and when cases were stratified by HR expression[16]. When looking at the frequency of gene alterations, no difference was found in triple-negative HER2-low vs. HER2-0 tumors, regardless of the stage, and no difference was found in early-stage hormone receptor (HR)-positive HER2-low vs. HER2-0 tumors; the only identified difference was a higher *TP53* mutational rate in metastatic HR-positive HER2-low vs. HER2-0 tumors (OR 1.49)[16]. Moreover, Schettini et al. found

only marginal differences between HER2-low and HER2-0 tumors at the transcriptomic level in a cohort of 1576 patients with HER2-negative breast cancer undergoing PAM50 profiling[17]. Beyond not differing in terms of molecular features, HER2-low tumors do not appear to show clinically relevant differences in terms of prognosis compared with HER2-0 tumors, as observed in a large cohort study from our group ($n = 5235$)[18] and in multiple similar studies[17,19–29]. Taken together, both molecular and prognostic studies suggest that HER2-low breast cancer does not represent a distinct molecular entity, but rather a heterogeneous group of tumors, whose biology and behavior is primarily dictated by HR expression. Of note, this view is consistent with the vote provided by 32 experts in the recently published ESMO Consensus Statements on HER2-low breast cancer.

Despite the lack of an obvious distinct genomic profile, HER2-low tumors evaluated in our study did show a significantly higher *ERBB2* copy number compared with HER2-0 tumors. Intriguingly, the frequency of *ERBB2* single-copy deletions among HER2-0 tumors was double that of HER2-low tumors (31.1% vs. 14.5%), an alteration that has been suggested to mediate resistance to T-DXd among a small subset (6%, $n = 5/88$) of patients treated in the DAISY phase 2 trial[30]. In this setting, the identification of a predictor of T-DXd activity in the HER2-negative disease remains critical, given that HER2 IHC scores appear inadequate for the purpose. Indeed, similar activity with T-DXd has been observed in patients having HER2 IHC 1+ and 2+/non-amplified tumors[6,11,31], and meaningful antitumor activity (overall response rate [ORR] 30%) was observed even in patients with HER2-0 tumors in the DAISY trial[32]. Exploration of different means to quantify HER2 status at the low end of the range and to predict the activity of T-DXd is thus urgently needed[33], with *ERBB2* copy counts being a promising biomarker warranting further study. Overall, the demonstration of the molecular similarity between HER2-low and HER2-0 breast tumors supports the notion that both subsets may potentially benefit from T-DXd. This finding is partly being tested in the DESTINY-Breast06 phase 3 trial, which tests T-DXd not only among patients with HER2-low but also with "ultralow" HER2 expression (i.e., >absence <1+, which is currently considered a subset of HER2-0)[34]. Nonetheless, until more prospective data are available, our analysis should not be extrapolated to treat patients with HER2-0 breast cancer with T-DXd.

Limitations of our study include the retrospective nature of the analysis, the inclusion of a mixture of samples obtained from metastatic biopsies and primary tumors, the inclusion of FPPE from biopsies that may not fully capture the spatial heterogeneity of the disease, the known temporal instability in HER2 expression, the heterogeneity of treatment administered to patients and the lack of study-specific central rescoring for HER2 IHC central determination of HER2 IHC scores. Nonetheless, all efforts were dedicated to minimizing the risk of bias. Indeed, the inclusion of a large sample size (>1000 patients) and the use of a consistent method of genomic analysis are expected to increase the reliability of the results. Moreover, sensitivity analyses were conducted to strengthen the study results: both when restricting the comparison at patients with metastatic disease at the time of sampling and when restricting the comparison at HER2 IHC 2+ vs. HER2-0 we obtained results that are consistent with the overall study cohort. Lastly, the comparable proportion of biopsies (vs. resections) utilized for conducting NGS between HER2-low and HER2-0 patients is expected to limit the impact of spatial heterogeneity on the results of our study, reducing the risk for bias.

In conclusion, among a large cohort of patients with HER2-negative MBC, the genomic landscape of HER2-low and HER2-0 tumors did not differ significantly after correcting for ER expression, apart from a higher average number of *ERBB2* alleles among HER2-low tumors. This study supports the notion that HER2-low tumors, as currently defined, should not be considered a distinct molecular subtype of breast cancer.

## Methods

### Ethical considerations

We analyzed clinicopathologic data from a prospectively maintained institutional database of patients with MBC treated at Dana-Farber Cancer Institute from July 1, 2013, to December 31, 2020, and provided written informed consent to DF/HCC IRB #11-104 and/or #17-000 (PROFILE study), which allows for genomic profiling with a targeted exome, tumor-only next-generation sequencing (NGS) platform (OncoPanel)[35]. The study was approved under a dedicated protocol (DF/HCC IRB #17-482) allowing for clinical annotation linked to OncoPanel data for patients with breast cancer and was carried out in accordance with the Declaration of Helsinki.

### Inclusion criteria

Patients were included if they were diagnosed with de novo or recurrent MBC, and if they had HER2-negative disease per ASCO/CAP criteria at the time of initial metastatic diagnosis or, if no metastatic diagnostic biopsy was performed, at the time of primary breast cancer diagnosis. Patients were excluded if no information on the HER2 IHC score could be retrieved; if IHC 2+ without available ISH; or if their primary tumor tested HER2-positive. We then identified patients who underwent successful NGS on a specimen with detailed HER2 status available. For those patients who received NGS on more than one specimen, we selected the closest tumor sample tested after the patient's initial metastatic diagnosis. Patients were classified as either HER2-low if they had a HER2 IHC score of 1+ or 2+ with negative ISH assay, or HER2-0 if they had an HER2 IHC score of 0 based on the tumor sample tested by NGS.

HER2 status, along with ER and progesterone receptor (PR) expression, were abstracted from pathology records. Tumors were considered HR-positive if at least 1% of invasive tumor cells exhibited immunostaining for either ER or PR[36]. Among HR-positive tumors, patients were further divided into ER-low (ER = 1–9%) and ER-positive (ER ≥ 10%). Other clinicopathologic parameters evaluated according to HER2-low or HER2-0 status were: age (date of birth), sex, and ethnicity group, which were self-reported and abstracted from the medical record based on the demographic information collected at the time of patient registration; stage, tumor histology, ER, PR, and HER2 status at primary diagnosis, metastatic diagnosis and for the sample tested by OncoPanel, timing of sample tested (e.g., primary, local recurrence or metastatic setting), number/type of metastatic sites, receipt of (neo)adjuvant therapy, and receipt of prior treatments in the metastatic setting.

### Tumor genomic analysis

Tumors for all MBC patients were assessed on a targeted, tumor-only NGS sequencing platform (DFCI-Profile) using the previously validated OncoPanel assay[37]. The assay surveys 277 (version 1), 302 (version 2), or 447 (version 3) cancer-associated genes. Genomic testing on formalin-fixed, paraffin-embedded (FFPE) tissue was performed centrally within the Center for Advanced Molecular Diagnostics at Brigham and Women's Hospital (Boston, MA), a Clinical Laboratory Improvement Amendments–certified laboratory environment, according to methods that were previously published[37]. All FFPE samples underwent histopathologic review prior to DNA extraction for the determination of adequacy and tumor cellularity, with selection of cases with at least 20% tumor cellularity. A manual review was also conducted for samples with no mutations, no record tumor purity, or pathology review indicating that the sample may have failed quality checks during sequencing, and samples that did not meet minimum sequencing depth standards were excluded. Standard methods were applied for DNA extraction[38]. Test results were reviewed by laboratory staff and interpreted by board-certified pathologists. Genomic calls included mutations and CNVs in genes that were only found common in all 3 versions of OncoPanel using the previously described pipelines[35]. Mutations were further filtered by removing variants present in the gnomAD[39] and ClinVar[40] databases (both the 7/31/19 update) unless they were found more than once in the

COSMIC[40] database (7/11/19 update) to remove non-cancer-causing germline mutations. Filtered mutations were classified as oncogenic using the OncoKB tool[41] (including the "Predicted Oncogenic" and "Likely Oncogenic" labels as oncogenic) and further in-house determination of loss-of-function (LOF) mutations in tumor suppressor genes (TSGs) (Supplementary Table 2). CNVs were further determined as oncogenic if they were called high amplifications (for OncoPanel, this refers to any amplification with greater than 6 counts) for oncogenes, or as a two-copy loss for TSGs.

TMB was estimated by counting all mutations after filtering by gnomAD and COSMIC and dividing by total panel size. Tumors were further classified as hypermutated (HM) using segmentation analysis[42] to determine an inflection point for the rapid increase in TMB; this was a TMB greater than 15 for this cohort.

*ERBB2* integer copy counts were calculated for tumors with recorded histology-estimated purities and copy number segmentation using a simple model of allelic gain/loss given by the formula $\frac{(2^{\text{Log2R ERBB2}}+1)-(2*(1-\text{Tumor Fraction})}{\text{Tumor Fraction}} \frac{(2^{\text{Log2R}ERBB2}+1)-(2*(1-\text{Tumor Fraction})}{\text{Tumor Fraction}}$ As OncoPanel does not sequence matched normal, these integer copy counts do not account for tumor ploidy and should be treated as relative copy counts to the overall tumor ploidy.

### Statistical analysis

Clinical and baseline genomic categorical characteristics were compared using chi-square between samples with HER2-low and HER2-0 status. Further comparison was done by stratified analysis on ER status, positive (here defined as ER ≥ 1%) vs. negative, using the Cochran-Mantel-Haenszel test. Continuous and nominal variables were compared using either a *t*-test or the Kruskal-Wallis test, and stratified values were calculated using the asymptotic stratified Kruskal-Wallis test found in the R package coin[43]. For statistical analyses or figures where ER status was treated as a covariate or stratifying factor, patients without recorded ER status were excluded (*n* = 6). Exact statistical procedures are described in the legend of each relevant figure.

Genomic event enrichment (mutations and CNVs) was determined by logistic regression models to account for background event rate and ER status. HM status was used in background modeling as a categorical variable as the number of oncogenic mutations increases logarithmically with total number of mutations (Supplementary Fig. 7). A similar value was used for copy number by setting a categorical cut-off on the number of high amplifications and deep deletions and based on thresholding and where the model performed the best. HER2 status was treated as a categorical variable. Models that did not converge after 500 iterations were excluded. *P*-values were calculated for the log-likelihood (LL) of each model and corrected for multiple hypothesis testing using Benjamini-Hochberg false discovery rate (BH-FDR) to produce corrected *p*-values (*q*-values). Those models that had LL *q*-value < 0.05 were included in the final results. *P*-values for coefficients were also corrected for multiple hypothesis testing using BH-FDR.

For comparing *ERBB2* copy number status, copy counts were converted to a categorical variable based on a simple interpretation of allelic copies with noise (single-copy deletion: <1.5 calculated copies, no copy variation: 1.5-2.5 copies, allelic gain: >2.5 copies), and HER2 status was considered as either quantitative (HER2 IHC 0, 1+, 2+/non-amplified) or categorical variable (HER2-low vs. HER2-0). Samples with a recorded IHC value of "Not Otherwise Specified" when under "2+" were excluded from any comparisons of *ERBB2* copy counts, as were those without recorded tumor purity or copy number segmentation intermediate files generated by the OncoPanel pipeline

### Sensitivity analysis

As the overall analysis was derived from patients who had either primary or metastatic disease at the time of collection of the tissue sample used for OncoPanel testing, we performed a sensitivity analysis

by restricting comparisons to only patients who had metastatic disease at the time of tissue sample collection and otherwise met the inclusion criteria. An additional sensitivity analysis was performed to compare the genomic landscape of IHC 2+ and IHC 0 tumors.

## Reporting summary

Further information on research design is available in the Nature Portfolio Reporting Summary linked to this article.

## Data availability

The genomic data generated in this study have been deposited in AACR Project GENIE cBioPortal under the GENIE cohort public project. The genomic data in GENIE are under restricted access that can be accessed when agreeing to the terms and conditions of GENIE use. Clinical molecular, histological, and staging datasets and internally filtered genomic datasets are available in the Source Data with matching GENIE identifiers. However, only GENIE data have exact mutational data such as base change. Exact mutations for samples missing from GENIE, as well as additional clinical information, may be acquired from the corresponding author P.T. (Paolo_Tarantino@dfci.harvard.edu) if the researcher has proper access as described in the study protocol and the patient has agreed to sharing data with external entities. The timescale for data to be made available is approximately 6 months and the data will be available for 3 years. Source data are provided with this paper.

## Code availability

Code is included for all analyses that required non-standard use of existing analysis packages; all other code for analysis and figures can be provided upon reasonable request to the corresponding author, P.T. (Paolo_Tarantino@dfci.harvard.edu). Requests will need to be supported by a description of the plan to utilize the code. The plan will be reviewed by the co-first authors and the senior authors, with the aim to provide the code within 6 months of the request.

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

## Acknowledgements

We thank Valerie Hope Goldstein, a full-time employee of Dana-Farber Cancer Institute, for submission assistance. This work was supported by AstraZeneca, which provided financial funding for this study (to P.T., grant number 5122601), but was not involved in the design, collection, management, analysis, and interpretation of the data. Article submission for publication was not dependent on approval from the funder. We also acknowledge support from NCI DF/HCC SPORE in Breast Cancer (P50 CA168504); the Breast Cancer Research Foundation (to N.U.L.), the Pan-Mass Challenge (to Dana-Farber Cancer Institute Breast Oncology Program), Fashion Footwear Association of New York (to Dana-Farber Cancer Institute Breast Oncology Program), National Comprehensive Cancer Network Oncology Research Program-Pfizer Independent Grants for Learning and Change (to N.U.L.), the Cross Family Fund for Triple-Negative Breast Cancer Research (to Dana-Farber Cancer Institute Breast Oncology Program), the Saverin Breast Cancer Research Fund (to Dana-Farber Cancer Institute Breast Oncology Program), and OOFOS (to Dana-Farber Cancer Institute Breast Oncology Program).

## Author contributions

Concept and design: P.T., H.G., S.M.T., N.U.L. Acquisition, analysis, or interpretation of data: P.T., H.G., M.E.H., J.F., S.S., G.K., A.-M.F., Y.L., A.C.G.-C., R.B.-S., B.L.B., S.D., L.S., L.M., N.L., B.E.J., M.M., R.J., X.Q., R.L., H.L., E.P.W., D.D., G.C., A.D.C., S.M.T., N.U.L. Drafting of the manuscript: P.T., H.G., M.E.H. Critical revision of the manuscript for important intellectual content: P.T., H.G., M.E.H., J.F., S.S., G.K., A.-M.F., Y.L., A.C.G.-C., R.B.-S., B.L.B., S.D., L.S., L.M., N.L., B.E.J., M.M., R.J., X.Q., R.L., H.L., E.P.W., D.D., G.C., A.D.C., S.M.T., N.U.L. Statistical analysis: H.G., A.D.C. Obtained funding: P.T., S.M.T. Administrative, technical, or material support: M.E.H. Supervision: S.M.T., N.U.L.

## Competing interests

P.T. served as advisor/consultant for AstraZeneca, Daiichi-Sankyo, Gilead and Lilly. N.U.L. reports institutional research support from Genentech, Pfizer, Merck, Seattle Genetics, Zion Pharmaceuticals, Olema Pharmaceuticals, and AstraZeneca; consulting honoraria from Puma, Seattle Genetics, Daiichi-Sankyo, AstraZeneca, Denali Therapeutics, Prelude Therapeutics, Olema Pharmaceuticals, Aleta Bio-Pharma, Affinia Therapeutics, Voyager Therapeutics, Janssen, Blueprint Medicines, Stemline/Menarini, and Artera Inc. and Reverie Labs;.; and royalties from UptoDate (book). S.M.T. reports consulting or advisory roles for Novartis, Pfizer, Merck, Eli Lilly, AstraZeneca, Genentech/Roche, Eisai, Sanofi, Bristol Myers Squibb, Seattle Genetics, CytomX Therapeutics, Daiichi-Sankyo, Gilead, Ellipses Pharma, 4D Pharma, OncoSec Medical Inc., BeyondSpring Pharmaceuticals, OncXerna, Zymeworks, Zentalis, Blueprint Medicines, Reveal Genomics, ARC Therapeutics, Infinity Therapeutics, Myovant, Zetagen, Umoja Biopharma, Artios Pharma, Menarini/Stemline, Aadi Biopharma, Bayer, Incyte Corp, and Jazz Pharmaceuticals; and research funding from Genentech/Roche, Merck, Exelixis, Pfizer, Lilly, Novartis, Bristol Myers Squibb, Eisai, AstraZeneca, Gilead, NanoString Technologies, Seattle Genetics, and OncoPep. G.C. reports honoraria for speaker's engagement from Roche, Seattle Genetics, Novartis, Lilly, Pfizer, Foundation Medicine, NanoString, Samsung, Celltrion, BMS, MSD; honoraria for providing consultancy from Roche, Seattle Genetics, NanoString; honoraria for participating on the advisory boards of Roche, Lilly, Pfizer, Foundation Medicine, Samsung, Celltrion, Mylan; honoraria for writing engagement from Novartis and BMS; honoraria for participation in the Ellipsis Scientific Affairs Group; institutional research funding for conducting phase I and II clinical trials from Pfizer, Roche, Novartis, Sanofi, Celgene, Servier, Orion, AstraZeneca, Seattle Genetics, AbbVie, Tesaro, BMS, Merck Serono, Merck Sharp Dome, Janssen-Cilag, Philogen, Bayer, Medivation, and Medimmune. A.C.G.-C. reports research funding (to Institution) from AstraZeneca, Daiichi-Sankyo, Merck, Gilead Sciences, Zenith Epigenetics; and travel accommodations from Roche/Genentech. R.B.-S. reports receiving speaker bureau fees from Agilant, AstraZeneca, Daiichi-Sankyo, Eli Lilly, Pfizer, Novartis, Merck, and Roche. He has also served as a consultant/advisor for AstraZeneca, Eli Lilly, Libbs, Roche, Merck and has received support for attending medical conferences from AstraZeneca, Roche, Eli Lilly, Daiichi-Sankyo, and Merck. D.D. has served as a consultant for Novartis, on the Advisory Board for Oncology Analytics and receives research funding from Canon Inc. B.E.J. reports that he has served as a paid consultant to Novartis, Checkpoint Therapeutics, Hummingbird Diagnostics, Daiichi- Sankyo, AstraZeneca, G1 Therapeutics, BlueDotBio, GSK, Hengrui Therapeutics, Simcere Pharmaceutical,

Jazz Pharmaceuticals, and Merus N.V. He is also an unpaid member of a Steering Committee for Pfizer. He receives research support from Cannon Medical Imaging. A.D.C. receives research support from Bayer AG. M.M. reports research support from Bayer AG and Janssen; consulting for Bayer, Delve, Interline, and Isabl; and royalties from Bayer and LabCorp. R.J. reports research support from Pfizer and Lilly and serves as an advisor to GE Health and Carrick Therapeutics. The remaining authors declare no competing interests.
