## [Peer Review File · Nature Communications]

Comprehensive genomic characterization of HER2-low and HER2-0 breast cancerREVIEWER COMMENTS

Reviewer #1 (Remarks to the Author): expertise in HER2-low breast cancer

The emergence of new antibody drug conjugates (ADCs) with demonstration of activity in HER2 low breast cancer revisited the binary definition of HER2 status positive (IHC3+or amplification) versus HER2 negative (HER2 0, 1+ or 2+ ISH negative). It was important to state if HER2 low breast cancer represents a new biological entity or if we should consider HER2 low status as “a biomarker”.

The aim of the study was to characterize the genomic profile of HER2 low tumors in a large population of MBC patients at Dana Farber Institute and to compare with the genomic profile of HER2-0 tumors.

The cohort is large with a total of 1039 patients.

The methods are well described. Important to note for ER status, the authors not only looked at positive versus negative according to ASCO guidelines but also ER-low (1-9%) which is considered as negative in Europe.

In this large cohort, the genomic landscape of HER2 low and HER2-0 does not differ significantly apart from a higher ERBB2 copy count among HER2-low tumors and higher rate of ERBB2 heterozygous deletions in HER2-0 tumors.

The data presented support the conclusion that HER2-low is not a molecular entity and are concordant with other genomic studies presented.

The limitations of the study are also well presented.

Minor comments

-the single-institution, retrospective nature of the analysis in my opinion it is not really a limitation considering the experience of the laboratory staff and board specified pathologists

- the inclusion of a mixture of samples obtained from metastatic biopsies and primary tumors yes indeed but a sensitivity analysis was performed looking to 801 samples collected during metastatic disease and this analysis is consistent with the overall analysis.

-the lack of central determination of 237 HER2 IHC scores not clear for me as in sentence 267 “Patients were classified as either HER2-low if they had a HER2 IHC score of 1+ or 2+ with 268 negative ISH assay, or HER2-0 if they had HER2 IHC score of 0 based on the tumor sample 269 tested by NGS”It was abstracted from the pathology report not reviewed ? We know there is a moderate concordance with local versus central testing for HER2 low (Fernandez *Al Jama Oncol* 2022).

- Moreover we know that HER2-low status is not biologically stable or consistent... but very difficult to assess in a homogeneous population.

Minor comments table 1. Some patients (few 2.3%) received prior neo/adjuvant anti HER2 treatment: They were considered as HER2 positive at initial diagnosis? Prior ADC received for metastatic disease at time of OncoPanel testing (0.7%). <Perhaps these cases should not be included in the analysis but according to numbers and same proportion in HER2-low and HER2-0, the results should be exactly the same.

This important genomic characterization reinforced the fact that HER2-low is not a new molecular entity: there is no significant differences in the incidence of oncogenic mutations after correcting for ER expression.

The higher ERBB2 copy count among HER2-low tumors should be explored in ongoing trials as a

potential biomarker.

Reviewer #2 (Remarks to the Author): expertise in breast cancer genomics

This manuscript by Tarantino and Colleagues details efforts to describe the genomic differences (if any) between HER2-low and HER2-0 breast cancer as defined by standard immunohistochemical methods. The authors present analysis performed using the DFCI's OncoPanel next generation sequencing test on a curated retrospective clinical dataset of over 1000 patients with clinically HER2-negative breast cancer. The authors find no genomic difference between these patient populations, with the exception that ERBB2 copy number appears lower in HER2-0 compared to HER2-low breast cancer. This finding is overall consistent with several other reports that HER2-low breast cancer does not represent a distinct biologic subtype.

Major comments/questions:

- What was the rationale for exclusion of patients with HER2+ primary who had HER2-negative MBC? It seems reasonable to include the patients with HER2- metastatic disease for whom the HER2- metastatic samples were sequenced. In Table 1, the authors report a few cases that had received (neo)adjuvant anti-Her2 therapy. Can the authors explain why this is the case if the HER2+ primaries had been excluded?

- Several prior reports including the recent study from the Yale group (Fernandez, et al, JAMA Onc 2022) have demonstrated poor reproducibility and accuracy of HER2 IHC readings, particularly when pathologists were asked to distinguish IHC 0 vs 1+. This issue is expected to have a much higher magnitude in the historic data (i.e. HER2 IHC status prior to approval of T-DXd such as the data presented in this manuscript) as distinguishing HER2 0 vs 1+ had no clinical significance and pathologists had no incentive to accurately distinguish between these 2 categories. This nondifferential misclassification of IHCs could significantly affect the power of the study in identifying the differences between genomic characteristics of HER2-0 vs HER2-low groups. Considering these issues, it is critical to repeat the analyses, comparing HER2-0 vs HER2-2+/ISH- categories that are less likely to be affected by IHC misclassification. It would be ideal to perform a pathology IHC review of at least a random subset of the samples included in the study to confirm the accuracy of the reported IHC results.

- The methods should be clarified surrounding the central finding of ERBB2 copy number changes. Currently the manuscript notes only that "ERBB2 copy counts were calculated for tumors with recorded histology-estimated purities and copy-number segmentation using a simple model of allelic gain/loss." What specific method does this assay use? ABSOLUTE? Were results normalized to tumor ploidy? Also, can the authors explain and justify the choice of threshold for copy number gain/loss of <1.5, 1.5-2.5 copies, and >2.5? Have these thresholds been validated within the OncoPanel Assay? Are these based on Log2R or total allelic ERBB2 copy numbers (TCN)?

- 3 different versions of OncoPanel have been utilized, are the analyses take into account the differences

between the bed files of the panels for these versions? How?

- In Figure 1A, the number of CNVs that are considered to be VUS are in order of magnitude higher than what is known to occur in breast cancer? How the CNV VUSs are identified and categorized?

- The legends often do not include the exact number of samples included in each analysis.

- A frequency threshold of 5% was indicated in the text for inclusion of CNVs in the analyses while a threshold of 4% was used for the mutations? What was the rationale for choosing different thresholds? The methods section also does not include these details.

- How did the authors deal with the low purity tumors affecting mutation and CNV calls? Are they excluded or included in the analyses? A total of 724 samples were included in the ERBB2 CNV analysis. What was the reason for excluding the other samples? This needs to be discussed in the methods.

Minor Comments/questions:

- Are there patients who had more than one tumor analyzed? How are the samples selected for this study if a patient has more than one sequenced tumor?

- Would suggest adding the result for ERBB2 allele difference (not just the p-value) to the abstract.

- Among patients without single copy deletions, did ERBB2 copy number still differ?

- Lines 128-129, what is meant by 'high' amplifications?

- Line 231, I would contend that ERBB2 copy counts may not be a good patient level biomarker overall, however it is possible that ERBB2 gene loss could predict poor response to T-DXd and act as a biomarker in this sense.

- Was homozygous loss of ERBB2 observed?

- Table 1, the race category "American Indian, Aleutian, Eskimo" has values of 0 across the categories and could be included in the "Other" category.

- Table 1, the frequency of DCIS was reported to be 15 under Stage at Dx and 17 under Histology at Dx. Which one is correct?

- Table 1, "Time from initial met diagnosis to OncoPanel test" was reported as median (min, and max). Would be more informative to report Median and IQRs. Are these values in days? Is this interval from the time that the sample was collected, or the test was ordered?

- The sensitivity analysis restricted to metastatic samples showed similar findings. However, TMB, FGA, ploidy, and mutational and copy number profiles are expected to be different comparing primary vs metastatic tumors, and this could have affected some of the findings in the overall analyses due to potential differential biases. Hence, it seems reasonable for the logistic regression models to be also adjusted for the type of sample sequenced.

- It is not clear how the samples in the Fig 2A oncoprint are sorted as the oncoprint starts and ends with samples that have no alterations in the genes included in the oncoprint.

- In Fig 3, the dots representing the ORs are too small to be appreciated.

Reviewer #3 (Remarks to the Author): expertise in bioinformatics analysis

This manuscript compared the genomic (mutations and CNVs on target panel genes) differences between HER2-low and HER2-0 breast cancer (mostly metastatic). No much difference was found except that HER2-low tumors had a higher number of ERBB2 compared to HER2-0. The study was well carried out and the manuscript is well written. However, most of these results are not unexpected. The information from targeted NGS is incomplete considering less than 500 genes are tested and there must be other genomic differences not captured. Some other minor comments:

- 1) For FFPE samples, are they mostly from biopsy or resection? The information is not provided but important as small sampling can not assess tumor heterogeneity.
- 2) It is not clear how CNV was called or detected so this should be provided in the method section.
- 3) What would be the clinical implications for the patient management from these findings? Should two groups be treated equally or is there still need to define the two groups?

Author Response Letter

We would like to thank our expert reviewers for providing valuable feedback on our work. We have now extensively revised the manuscript to address all the comments and requests we received.

In particular, to address the concerns of Reviewer #2 regarding the IHC classification used, we performed a new sensitivity analysis comparing HER2-0 with HER2-2+, which ultimately confirmed and strengthened the overall results of our work. Moreover, we have addressed the concerns of Reviewer #3 by providing granular data on the types of samples used (which were comparable between the two groups) and by providing detailed explanation on the rationale behind the methods utilized in our work.

We believe that these revisions have improved the strength and clarity of our findings, and we are grateful to the reviewers for taking the time to provide these helpful suggestions.

REVIEWER COMMENTS

Reviewer #1: expertise in HER2-low breast cancer

Reviewer #1: The emergence of new antibody drug conjugates (ADCs) with demonstration of activity in HER2 low breast cancer revisited the binary definition of HER2 status positive (IHC3+or amplification) versus HER2 negative (HER2 0, 1+ or 2+ ISH negative). It was important to state if HER2 low breast cancer represents a new biological entity or if we should consider HER2 low status as “a biomarker”. The aim of the study was to characterize the genomic profile of HER2 low tumors in a large population of MBC patients at Dana Farber Institute and to compare with the genomic profile of HER2-0 tumors. The cohort is large with a total of 1039 patients. The methods are well described. Important to note for ER status, the authors not only looked at positive versus negative according to ASCO guidelines but also ER-low (1-9%) which is considered as negative in Europe. In this large cohort, the genomic landscape of HER2 low and HER2-0 does not differ significantly apart from a higher ERBB2 copy count among HER2-low tumors and higher rate of ERBB2 heterozygous deletions in HER2-0 tumors. The data presented support the conclusion that HER2-low is not a molecular entity and are concordant with other genomic studies presented. The limitations of the study are also well presented.

Authors: We would like to thank Reviewer #1 for nicely describing the aims and results of our analysis, and for the kind words on this study.

Minor comments

Reviewer #1: the single-institution, retrospective nature of the analysis in my opinion it is not really a limitation considering the experience of the laboratory staff and board specified pathologists

- **Authors:** We thank Reviewer 1 for this consideration. We agree that, given the established expertise of the pathologists and laboratory staff involved in the study, it could be argued that the single-institution nature of the study may be a strength rather than a limitation. We have thus removed this language from the limitations listed in the discussion.

Change: removed "single-institution" (page 13 of the revised tracked edits manuscript)

Reviewer #1: the inclusion of a mixture of samples obtained from metastatic biopsies and primary tumors yes indeed but a sensitivity analysis was performed looking to 801 samples collected during metastatic disease and this analysis is consistent with the overall analysis.

- **Authors:** We also thank Reviewer 1 for this consideration. We did our best to limit any bias associated with our retrospective analysis, including performing the sensitivity analysis which, as well detailed by Reviewer 1, confirmed the overall results of the study.

Reviewer #1: the lack of central determination of 237 HER2 IHC scores not clear for me as in sentence 267 "Patients were classified as either HER2-low if they had a HER2 IHC score of 1+ or 2+ with 268 negative ISH assay, or HER2-0 if they had HER2 IHC score of 0 based on the tumor sample 269 tested by NGS"....It was abstracted from the pathology report not reviewed? We know there is a moderate concordance with local versus central testing for HER2 low (Fernandez *Al Jama Oncol* 2022).

- **Authors:** We realize that concordance between local and central testing for HER2 IHC is not ideal, although more recent studies (e.g. by Viale G. et al. *ESMO Open* 2023 and Ruschoff J. et al. *SABCS22*) have consistently found >80% concordance between HER2-0 and HER2-low, which is much higher than that reported by Fernandez et al, and overall reassuring. Additionally, the practice-changing study of T-DXd in HER2-low breast cancer (*DESTINY-Breast04*) reported approximately 80% concordance between HER2-0 and HER2-low (Prat A. et al *SABCS22*). It would not be feasible to re-request blocks for the more than 1,000 patients included in this study to perform central HER2 testing, particularly as many patients had their biopsy or surgery outside of our institution.

Of note, as part of standard institutional practice, outside HER2 IHC slides are routinely requested, and are reviewed when available, by a dedicated breast pathologist as part of usual medical care for patients coming to our institution for a second opinion and who had tissue sampling outside of our institution. If there is clinical concern about the validity of the HER2 IHC slides (e.g. due to overstaining of slides, unusual clinical history, etc.), then blocks are requested

in real time and HER2 stains are repeated at our institution as part of routine clinical care. However, once slides have been reviewed, they are sent back to the originating institution.

We recognize that the ideal case would be to conduct central HER2 testing including with standard antibody-based IHC assays, and potentially even with the novel, more quantitative HER2 IHC assays using antibodies more sensitive to the low HER2 range; however, for the reasons above, this is not feasible, and would be very costly for such a large cohort of patients. Additionally, we believe that our results reflect HER2-low classification as utilized in routine patient care. We have included these limitations in our revised Discussion.

Reviewer #1: Moreover we know that HER2-low status is not biologically stable or consistent... but very difficult to assess in a homogeneous population.

- **Authors:** We absolutely agree with this consideration, which we have also previously found in a dedicated study (Tarantino et al. Eur J Cancer 2022). Given the high instability of HER2-low expression, we decided to determine the HER2 status of each patient according to the tissue sample that received NGS testing, so that we could be sure of the attribution of each genomic finding to the right HER2-status. We have now mentioned this aspect among the limitations of the study.

Added sentence: "the known temporal instability in HER2-expression" was added among the study limitations (page 13 of the revised tracked edits manuscript)

Reviewer #1: Minor comments table 1. Some patients (few 2.3%) received prior neo/adjuvant anti HER2 treatment: They were considered as HER2 positive at initial diagnosis?

- **Authors:** We realize that a very small number of patients included in the study has received prior HER2-targeted treatments. This aspect is related to the participation of patients in clinical trials or to complexities in HER2 assessment. In particular, two patients had been enrolled in the NSABP B-47 trial, which tested the role of adjuvant trastuzumab for HER2-negative (HER2-low) breast cancer. Most other cases received anti-HER2 treatments at outside centers, but were found to have HER2-negative disease upon the DFCI review of pathology, and thus were considered HER2-negative for the purpose of this analysis. Lastly, some cases had heterogeneity in HER2 IHC status that led to anti-HER2 treatment, but were ultimately found to be HER2-negative on FISH testing. Overall, we favor defining the above patients as HER2-negative, and we do not expect this small proportion of patients to relevantly impact the results of the study.

Reviewer #1: Prior ADC received for metastatic disease at time of OncoPanel testing (0.7%). Perhaps these cases should not be included in the analysis but according to numbers and same

proportion in HER2-low and HER2-0, the results should be exactly the same.

- **Authors:** Given the similarities of ADCs with chemotherapy treatment, leading to uncertain influence on the tumor genomic status, as well as the small number of these patients and the balance between arms (acknowledged by Reviewer 1), we ultimately decided to include these few patients in the analysis.

Reviewer #1: This important genomic characterization reinforced the fact that HER2-low is not a new molecular entity: there is no significant differences in the incidence of oncogenic mutations after correcting for ER expression. The higher ERBB2 copy count among HER2-low tumors should be explored in ongoing trials as a potential biomarker.

- **Authors:** we thank Reviewer 1 for this kind comment. We agree that, according to these data, HER2-low does not appear to be a distinct molecular entity. This is consistent with the choice by the 32 experts included in the ESMO Consensus on HER2-low breast cancer, recently published in *Annals of Oncology* and that we have now added as a reference to the manuscript.

Reviewer #2: expertise in breast cancer genomics

Reviewer #2: This manuscript by Tarantino and Colleagues details efforts to describe the genomic differences (if any) between HER2-low and HER2-0 breast cancer as defined by standard immunohistochemical methods. The authors present analysis performed using the DFCI's OncoPanel next generation sequencing test on a curated retrospective clinical dataset of over 1000 patients with clinically HER2-negative breast cancer. The authors find no genomic difference between these patient populations, with the exception that ERBB2 copy number appears lower in HER2-0 compared to HER2-low breast cancer. This finding is overall consistent with several other reports that HER2-low breast cancer does not represent a distinct biologic subtype.

- **Authors:** We would like to thank Reviewer #2 for the nice synthesis of our study aims and results.

Major comments/questions:

Reviewer #2: What was the rationale for exclusion of patients with HER2+ primary who had HER2-negative MBC? It seems reasonable to include the patients with HER2- metastatic disease for whom the HER2- metastatic samples were sequenced. In Table 1, the authors report a few cases that had received (neo)adjuvant anti-Her2 therapy. Can the authors explain why this is the case if the HER2+ primaries had been excluded?

- **Authors:** We thank Reviewer 2 for raising this point. The main rationale to exclude patients with a prior sample testing HER2-positive was to mirror the inclusion criteria of DESTINY-Breast04, which established the role of HER2-low breast cancer in clinical practice, and which excluded patients with a history of HER2-positive disease. The presence of few patients with receipt of prior (neo)adjuvant anti-HER2 therapy is related to the participation in clinical trials (i.e. NSABP B-47, which exposed patients with HER2-negative disease to trastuzumab), the finding of HER2-negative disease upon review of pathology conducted at DFCI (despite exposure to HER2-targeted treatment at external sites), or the presence of HER2 heterogeneity with a negative FISH test. Nonetheless, the small number of these patients is not expected to influence our analyses.

Reviewer #2: Several prior reports including the recent study from the Yale group (Fernandez, et al, JAMA Onc 2022) have demonstrated poor reproducibility and accuracy of HER2 IHC readings, particularly when pathologists were asked to distinguish IHC 0 vs 1+. This issue is expected to have a much higher magnitude in the historic data (i.e. HER2 IHC status prior to approval of T-DXd such as the data presented in this manuscript) as distinguishing HER2 0 vs 1+ had no clinical significance and pathologists had no incentive to accurately distinguish between these 2 categories. This nondifferential misclassification of IHCs could significantly affect the power of the study in identifying the differences between genomic characteristics of HER2-0 vs HER2-low groups. Considering these issues, it is critical to repeat the analyses, comparing HER2-0 vs HER2-2+/ISH- categories that are less likely to be affected by IHC misclassification. It would be ideal to perform a pathology IHC review of at least a random subset of the samples included in the study to confirm the accuracy of the reported IHC results.

- **Authors:** We agree that, at present, the accuracy of pathologists reads of HER2 IHC slides in the low range (0-1+-2+) is far from ideal, for which reason we have included this aspect among the relevant limitations of our study on page 13 of the revised tracked edits manuscript. At the same time, more recent studies (e.g. by Viale G. et al. ESMO Open 2023 and Ruschoff J. et al. SABCS22) have consistently found 80% concordance between HER2-0 and HER2-low, which is much higher than that reported by Fernandez et al, and overall satisfying. Similarly, the practice-changing study of T-DXd in HER2-low breast cancer (DESTINY-Breast04) has found approximately 80% concordance between HER2-0 and HER2-low (Prat A. et al SABCS22). Moreover, in line with the recent ESMO Consensus statements, treatment decisions in clinical practice (e.g. the choice to use or not T-DXd for HER2-low disease) are currently based on historical scorings of HER2, since re-staining thousands of samples from patients is not expected to be practical; accordingly, our choice to utilize local reads would be consistent with the current clinical use of HER2-low as a biomarker. Lastly, the near totality of the slides were read by specialized breast cancer pathologists at a reference center (Brigham and Women's Hospital; see response to Reviewer 1 for additional details), a factor which is expected to increase the reliability of the results.

While we do believe that our methodology is consistent with current clinical practice, we agree that a comparison of HER2-0 with HER2 2+/ISH- can further strengthen the conclusions of the paper, and we truly appreciate this suggestion by the expert reviewer. We have now performed

a sensitivity analysis using the IHC 0 and IHC 2+/ISH- as the comparators (n=759, including 549 HER2-0 and 210 HER2 2+/ISH-). Consistent with the overall findings of our study, this sensitivity analysis found no significant differences in TMB, oncogenic mutations and copy number variations between the two groups, reinforcing the idea that these do not represent two distinct molecular entities. The results are presented on pages 10-11 of the revised tracked edits manuscript, and dedicated figures reporting this sensitivity analysis were added in the supplement.

Added (page 10): Given some evidence of limited concordance in the pathologic scoring of HER2 IHC 0 vs 1+, a sensitivity analysis was carried out by comparing only HER2-low designated samples that stained IHC 2+/not amplified against the HER2-0 cohort for samples with a recorded ER status. This filtering left a total of 759 samples (549 HER2-0, 210 HER2-low) for analysis.

Slightly more HER2 IHC 2+ samples were ER-positive (78.1%, $P < 0.001$) in this smaller cohort, while TMB was not significant between the two groups with HER2 IHC 2+ median TMB of 7.26 (0.76 – 38.02) and HER2-0 with a median TMB of 7.60 (0.0 – 111.36, $P = 0.86$). Common oncogenic genomic mutations in the HER2 IHC 2+ cohort included the same as in the larger analysis with PIK3CA (36.2% HER2 IHC 2+ samples; 26.3% ER negative, 37.5% ER positive-low, 38.7% ER positive), TP53 (32.9%; 78.9%, 87.5%, 19.6%), CDH1 (14.8%; 7.9%, 12.5%, 16.6%), GATA3 (13.8%; 0%, 0%, 17.8%), and ESR1 (11.4%; 0%, 12.5%, 14.1%) were the most common, which was similar to the overall cohort, though TP53 was replaced by PIK3CA as the most common mutation. Copy number events were also similar and a plot of commonly altered samples can be seen in Supplementary Fig. S6.

When comparing the mutational landscape and copy number landscape, no genes rose to significance when accounting for multiple hypothesis testing (Supplementary Fig. S6 and S7). Mutations in similar genes, TP53 and NF1, were found to be enriched in HER2-0 before multiple hypothesis testing. As the results of comparing the HER2 IHC and ERBB2 values was already described, the values are not restated here.

Reviewer #2: The methods should be clarified surrounding the central finding of ERBB2 copy number changes. Currently the manuscript notes only that “ERBB2 copy counts were calculated for tumors with recorded histology-estimated purities and copy-number segmentation using a simple model of allelic gain/loss.” What specific method does this assay use? ABSOLUTE? Were results normalized to tumor ploidy? Also, can the authors explain and justify the choice of threshold for copy number gain/loss of <1.5, 1.5-2.5 copies, and >2.5? Have these thresholds been validated within the OncoPanel Assay? Are these based on Log2R or total allelic ERBB2 copy numbers (TCN)?

- **Authors:** We appreciate the questions by the reviewer. As mentioned in the paper, a model that takes into account copy segmentation using the Log2R was used. The specific method is the formula $(2^{**}(\text{HER2_LR}+1)-(2*(1-\text{Frac_Tumor})))/(\text{Frac_Tumor})$, which takes into account the

base tumor purity. As for normalizing tumor ploidy and methods like ABSOLUTE, a limitation of OncoPanel is the lack of normal tissue sequencing for samples, which prevents accounting for tumor ploidy or the use of those methods. The thresholds that were selected (<1.5, 1.5-2.5 copies, and >2.5) are based on simple interpretation of allelic copy number being gained, staying neutral, or lost in the background of noisy Log2R and tumor purity estimates.

We have now revised the methods to better explain the above limitations of OncoPanel testing and to clarify what strategies were implemented for the analysis of copy number changes.

Added (page 17 of the revised tracked edits manuscript): a simple model of allelic gain/loss given by the formula $\frac{(2^{\text{Log2R}_{ERBB2}+1})-(2(1-\text{Tumor Fraction}))}{\text{Tumor Fraction}}$. As OncoPanel does not sequence matched normal, these integer copy counts do not account for tumor ploidy and should be treated as relative copy counts to the overall tumor ploidy.*

(page 18): copy counts were converted to a categorical variable based on a simple interpretation of allelic copies with noise

Reviewer #2: 3 different versions of OncoPanel have been utilized, are the analyses take into account the differences between the bed files of the panels for these versions? How?

- **Authors:** We appreciate the question by Reviewer 2. To account for the differences in OncoPanel versions utilized for this study, we decided to include only genes that were common in all three version. For ERBB2 specifically, which is the only gene where the actual number of bases in each panel version matters, each panel sequenced the exact same range. We have revised the manuscript to clarify these aspects.

Added (page 16): Genomic calls included mutations and CNVs in genes that were only found common in all 3 versions of OncoPanel

Reviewer #2: In Figure 1A, the number of CNVs that are considered to be VUS are in order of magnitude higher than what is known to occur in breast cancer? How the CNV VUSs are identified and categorized?

- **Authors:** We thank Reviewer 2 for this comment and question. As mentioned in the legend, the unshaded bars that are labeled as VUS represent any other copy number event. Thus, these include any 1 copy deletion, any amplification with less than 6 copies, or a 2-copy deletion or higher amplification that occurs in an oncogene or tumor suppressor gene, respectively. We have revised the manuscript to clarify the broad definition of VUS, which explains the different rate compared to other studies.

*Added (pages 6-7): As any CNV event that were not high amplifications for oncogenes and 2-copy deletions for tumor suppressor genes were included in the “variants of unknown significance (VUS)” category, a significantly higher number of VUSs appear in **Fig. 1** as compared to other studies.*

Reviewer #2: The legends often do not include the exact number of samples included in each analysis.

- **Authors:** We are grateful to Reviewer 2 for raising this important point. To improve the clarity of the analyses, we have now added sample counts in the legend of each figure.

Reviewer #2: A frequency threshold of 5% was indicated in the text for inclusion of CNVs in the analyses while a threshold of 4% was used for the mutations? What was the rationale for choosing different thresholds? The methods section also does not include these details.

- **Authors:** We appreciate the reviewer asking for clarification on this point. The 4% threshold for mutations was purely adopted for visualization purposes in the figure; we have now revised the manuscript to clarify this aspect. No frequency threshold was indicated for inclusion of CNVs, and all of the converged model data, without any threshold, is presented in the Supplementary Table 1.

Added (page 7): only including common mutations for simpler visualization

Reviewer #2: How did the authors deal with the low purity tumors affecting mutation and CNV calls? Are they excluded or included in the analyses? A total of 724 samples were included in the ERBB2 CNV analysis. What was the reason for excluding the other samples? This needs to be discussed in the methods.

- **Authors:** Thank you for the comments. Low purity samples (<20% tumor content) were excluded from this work. Those that did not have a tumor purity reported by pathologists were manually examined to see if they had valid numbers of mutations, CNVs, and if molecular pathologists had commented on a lack of purity for exclusion. Only 724 samples were included in the ERBB2 CNV analysis as a number of the samples had an IHC that was not specified as “0” or “1+” but “0/1+ NOS”, some were missing histology estimated tumor purities, and a number of intermediate files that were used to calculate the copy counts could not be found from the original pipeline that all OncoPanel samples are through. We have now revised the methods to better clarify the selection criteria pertaining to tumor purity.

Added (page 16 of the revised tracked edits manuscript): Manual review was also conducted for samples with no mutations, no recorded tumor purity, or pathology review indicating that the sample may have failed quality checks during sequencing, and samples that did not meet minimum sequencing depth standards were excluded.

(page 18): Samples with a recorded IHC value of “Not Otherwise Specified” when under “2+” were excluded from any comparisons of ERBB2 copy counts, as were those without recorded tumor purity or copy number segmentation intermediate files generated by the OncoPanel pipeline

Minor Comments/questions:

Reviewer #2: Are there patients who had more than one tumor analyzed? How are the samples selected for this study if a patient has more than one sequenced tumor?

- **Authors:** Patients may have more than one tumor sequenced with OncoPanel over time during their care at DFCI. However, for the purpose of this analysis, we identified the first/closest tumor sample tested after the patient’s initial metastatic diagnosis. This aspect has been now clarified in the methods of the manuscript.

Added (page 14): For those patients that received NGS on more than one specimen, we selected the closest tumor sample tested after the patient’s initial metastatic diagnosis.

Reviewer #2: Would suggest adding the result for ERBB2 allele difference (not just the p-value) to the abstract.

- **Authors:** We are thankful for this suggestion. We have now added this value to the abstract.

Added (page 3): significantly higher number of ERBB2 alleles (2.05 median copy count) was observed among HER2-low tumors compared to HER2-0 (1.79 median copy count; p<0.001),

Reviewer #2: Among patients without single copy deletions, did ERBB2 copy number still differ?

- **Authors:** We thank the reviewer for the question. After removing the samples with single copy deletion, we found that the total copy count is still significantly different between the low and 0 category.

Reviewer #2: Lines 128-129, what is meant by ‘high’ amplifications?

- **Authors:** We appreciate the reviewer’s comment. High amplification refers in this paper to any gene with a copy count of 6 or above. We have revised the methods of the manuscript to clarify this definition.

Added (page 16): CNVs were further determined as oncogenic if they were called as high amplifications (*for OncoPanel, this refers to any amplification with greater than 6 counts*)

Reviewer #2: Line 231, I would contend that ERBB2 copy counts may not be a good patient level biomarker overall, however it is possible that ERBB2 gene loss could predict poor response to T-DXd and act as a biomarker in this sense.

- **Authors:** We agree with Reviewer 2 regarding the potential role of ERBB2 gene loss as a biomarker of resistance to T-DXd. As included in the discussion, this has been observed in a preliminary analysis of the DAISY study, and we are looking at this question in a separate analysis focusing on patients’ outcomes.

Reviewer #2: Was homozygous loss of ERBB2 observed?

- **Authors:** We thank Reviewer 2 for the question. We confirm that no homozygous loss of ERBB2 was observed. We have now included this finding in the manuscript.

Added (page 7 of the revised tracked edits manuscript): Of note, no homozygous loss of ERBB2 was observed in this cohort.

Reviewer #2: Table 1, the race category “American Indian, Aleutian, Eskimo” has values of 0 across the categories and could be included in the “Other” category.

- **Authors:** Thanks for the suggestion. We have removed this row from the table (i.e. now included among Others)

Reviewer #2: Table 1, the frequency of DCIS was reported to be 15 under Stage at Dx and 17 under Histology at Dx. Which one is correct?

- **Authors:** Thank you for noticing this discrepancy. We have now reviewed the stage at diagnosis for the 2 patients with DCIS histology, realizing that the Stage at Diagnosis should have been coded as 0/DCIS. We have fixed this in Table 1 of the revised manuscript.

Reviewer #2: Table 1, “Time from initial met diagnosis to OncoPanel test” was reported as median (min, and max). Would be more informative to report Median and IQRs. Are these values in days? Is this interval from the time that the sample was collected, or the test was ordered?

- **Authors:** We thank Reviewer 2 for the suggestion. The variable is calculated as the interval from the date of initial metastatic diagnosis to the date that the sequenced tumor sample was collected (e.g. date of biopsy/resection performed). As suggested, we have now added the median and interquartile ranges for this interval to the manuscript.

Reviewer #2: The sensitivity analysis restricted to metastatic samples showed similar findings. However, TMB, FGA, ploidy, and mutational and copy number profiles are expected to be different comparing primary vs metastatic tumors, and this could have affected some of the findings in the overall analyses due to potential differential biases. Hence, it seems reasonable for the logistic regression models to be also adjusted for the type of sample sequenced.

- **Authors:** We agree with the reviewer that there may be potential bias from differences in primary tumors, as we have established that there is none with metastatic tumors. For the sake of completeness we decided to rerun the models, which nonetheless left the results of the analysis unchanged.

Added (page 7): Reworded entire paragraph under “**Comparison of genomic profile by HER2 status**”; Figure 3 redone.

Reviewer #2: It is not clear how the samples in the Fig 2A oncoprint are sorted as the oncoprint starts and ends with samples that have no alterations in the genes included in the oncoprint.

- **Authors:** We thank the reviewer for noticing this issue. On inspection, we confirm that the OncoPrint was incorrectly formatted during post-production of figures. We have revised Figure 2 to fix the sorting of samples and genes.

Reviewer #2: In Fig 3, the dots representing the ORs are too small to be appreciated.

- **Authors:** We appreciate the input by Reviewer 2. We have increased the size of the dots in all figures to improve the visualization.

Reviewer #3: expertise in bioinformatics analysis

Reviewer #3: This manuscript compared the genomic (mutations and CNVs on target panel genes) differences between HER2-low and HER2-0 breast cancer (mostly metastatic). No much difference was found except that HER2-low tumors had a higher number of ERBB2 compared to HER2-0. The study was well carried out and the manuscript is well written. However, most of these results are not unexpected.

- **Authors:** We thank Reviewer #3 for the kind words on the conduction of our study and writing of the manuscript. We believe that demonstrating lack of significant molecular differences between HER2-low and HER2-0 tumors will provide helpful data to the discussion regarding the biologic bases of HER2-low expression.

Reviewer #3: The information from targeted NGS is incomplete considering less than 500 genes are tested and there must be other genomic differences not captured.

- **Authors:** We agree with Reviewer 3 that our targeted NGS panel does not cover all the potentially altered genes which could have been captured by WES/WGS. Nonetheless, OncoPanel does cover all the most commonly altered and oncogenically relevant genes that have been characterized to date; in this setting, further expanding the analysis to rare and less relevant genes is not expected to modify the results of our study. Moreover, our analysis reflects the real-world use of genomic profiling: indeed, all the commercially available genomic panels utilizing targeted sequencing rather than WES/WGS. Lastly, our methods are consistent with those of all other groups that have attempted answering this research question, including Marra et al.¹, Bansal et al.² and Berrino et al.³, further supporting the idea that the research community considers appropriate to utilize a targeted NGS panel to investigate the molecular underpinnings of HER2-low vs. HER2-0 tumors.

References:

- 1 Marra, A., Safonov, A., Drago, J., Ferraro, E. & Selenica, P. Genomic Characterization of Primary and Metastatic HER2-low Breast Cancers. *San Antonio Breast Cancer Symposium* (2022).

- 2 Bansal, R., McGrath, J., Walker, P., Bustos, M. A. & Rodriguez, E. Genomic and Transcriptomic Landscape of HER2-Low Breast Cancer. *San Antonio Breast Cancer Symposium* (2022).
- 3 Berrino, E. *et al.* Integrative genomic and transcriptomic analyses illuminate the ontology of HER2-low breast carcinomas. *Genome Medicine* **14**, 98 (2022).
<https://doi.org:10.1186/s13073-022-01104-z>

Some other minor comments:

Reviewer #3: For FFPE samples, are they mostly from biopsy or resection? The information is not provided but important as small sampling cannot assess tumor heterogeneity.

- **Authors:** We appreciate the comment by Reviewer #3. To address the question, we have now reviewed the procedures performed to collect tissue among the patients included in this study. As expected, most of the tissue derived from tumor biopsies performed in clinical practice for the biologic characterization of tumors (72%, n= 750). Importantly, the proportion of samples collected via biopsy is similar across HER2-low (71%, n= 344) and HER2 0 patients (73%, n=406). Therefore, any bias related to the small sampling of biopsies would be expected to be the same in both groups, and not ultimately affect the results of our analyses.

We have included the full table with the sample types below, and reviewed the manuscript to include the issue highlighted by Reviewer #3 among the limitations of the study.

All	Biopsy	Resection	Other	Total
Primary	89	147	0	238
Local/Regional Recurrence	14	10	0	24
Metastatic	647	111	20	777
Total	750(72%)	269(26%)	20(2%)	1039

HER2 Low	Biopsy	Resection	Other	Total
Primary	40	73	0	113
Local/Regional Recurrence	6	1	0	8
Metastatic	298	56	13	366
Total	344(71%)	130(27%)	13(3%)	487

HER2 0	Biopsy	Resection	Other	Total
Primary	50	75	0	125
Local/Regional Recurrence	7	9	0	16
Metastatic	349	56	6	411
Total	406 (73%)	139(25%)	7(1%)	552

Added (page 5): *In terms of procedures utilized to collect tissue, most of the tissue derived from tumor biopsies performed in clinical practice for the biologic characterization of tumors (72%, n=750), with a similar proportion of samples collected via biopsy across HER2-low (71%, n=344) and HER2-0 patients (73%, n=406).*

Added (page 13): *Limitations of our study include the retrospective nature of the analysis, the inclusion of a mixture of samples obtained from metastatic biopsies and primary tumors, the inclusion of FPPE from biopsies that may not fully capture the spatial heterogeneity of the disease...*

Reviewer #3: It is not clear how CNV was called or detected so this should be provided in the method section.

- **Authors:** We thank the reviewer for the comment. As mentioned in the manuscript, the OncoPanel pipeline that has been previously described was used to call CNVs and is provided below for information. Language has been included in the Methods to make more explicit how the calls were made, though for brevity's sake, the reader will be directed to the original OncoPanel paper.

“For copy number analysis, a custom R-based tool (VisCapCancer) was used to calculate the fractional coverage of specified genomic intervals compared with the median fractional coverage obtained in a panel of 67 FFPE nonneoplastic samples. Coverage across each interval captured was calculated using the “DepthOfCoverage” program of GATK.¹⁸ The final outputs of the tool were log₂ ratio values, which were plotted by relative genome order. All VisCap copy number plots were manually reviewed and interpreted with consideration of tumor percentage from initial pathologist review.”

Reviewer #3: What would be the clinical implications for the patient management from these findings? Should two groups be treated equally or is there still need to define the two groups?

- **Authors:** we thank Reviewer 3 for raising this important point. We believe that our data bring valuable insights regarding the biology of HER2-low breast cancer, but these results do not warrant any immediate change in its clinical management. Indeed, although HER2-low and HER2-0 breast cancer do not appear to be different on a genomic level, a benefit with T-DXd has been demonstrated in randomized trials only for patients with HER2-low disease, thus only HER2-low patients should receive T-DXd in clinical practice at the moment. At the same time,

the DAISY trial reported benefit of T-DXd in a small cohort of patients with HER2 0 disease. Our data support the notion that T-DXd should be further tested in patients with HER2-0 disease, as there appear to be no major genomic differences between HER2-low and HER2 0 breast cancer, and it is possible that even every low levels of HER2 expression (such that the tumor would currently be classified as HER2 0) are sufficient to drive clinical activity. In addition, our data support the concept that pharmacologic methods to increase the expression of HER2 in HER2 0 tumors might also increase activity of T-DXd, given that there are otherwise no inherent genomic differences between the two groups.

We have expanded the discussion to mention the impact of our study on the clinical management of HER2-low and HER2-0 breast cancer.

Added section (Page 13 of the revised tracked edits manuscript): “Overall, the demonstration of the molecular similarity between HER2-low and HER2-0 breast tumors supports the notion that both subsets may potentially benefit from T-DXd. This finding is partly being tested in the DESTINY-Breast06 phase 3 trial, which tests T-DXd not only among patients with HER2-low, but also with “ultralow” HER2 expression (i.e. >absence 0<1+, which is currently considered a subset of HER2-0)³⁴. Nonetheless, until more prospective data is available, our analysis should not be extrapolated to treat patients with HER2-0 breast cancer with T-DXd.”

REVIEWERS' COMMENTS

Reviewer #2 (Remarks to the Author):

I would like to thank the authors for their detailed response to my comments, all of which have been adequately addressed. The manuscript has significantly improved and I have no further comments.

REVIEWERS' COMMENTS

Reviewer #2 (Remarks to the Author):

I would like to thank the authors for their detailed response to my comments, all of which have been adequately addressed. The manuscript has significantly improved and I have no further comments.

Response: Our sincerest thanks to Reviewer #2 for helping us to enhance and improve our manuscript.